# IS IT NECESSARY TO APPLY PARAMETER-EFFICIENT FINE-TUNING TO ALL POSITION INDICES?

## ABSTRACT

In the field of large models (LMs) for natural language processing (NLP) and computer vision (CV), Parameter-Efficient Fine-Tuning (PEFT) has emerged as a resource-efficient method that modifies a limited number of parameters while keeping the pretrained weights fixed. This paper investigates the traditional PEFT approach, which applies modifications to all position indices, and doubts its necessity. We introduce a new paradigm called PEFT_S, in which a function S selectively applies PEFT modifications to a subset of position indices, potentially enhancing performance on downstream tasks. Our experimental results reveal that the indiscriminate application of PEFT to all indices is not only superfluous, but may also be counterproductive. This study offers a fresh perspective on PEFT, advocating for a more targeted approach to modifications and providing a framework for future research to optimize the fine-tuning process for large models.

## 1 INTRODUCTION

In recent research, large models (LMs) have become increasingly attractive in natural language processing (NLP)(Dubey et al., 2024; Yao et al., 2024), computer vision (CV)(Croitoru et al., 2023; Liu et al., 2024d;a), and other fields. Parameter-Efficient Fine-Tuning (PEFT) is a popular, resource-efficient fine-tuning approach that involves freezing pretrained weights while tuning only a limited number of new parameters(Hu et al., 2023; Han et al., 2024). Several PEFT methods, such as Prefix-Tuning(Li & Liang, 2021), Prompt-Tuning(Lester et al., 2021), BitFit(Zaken et al., 2022), LoRA(Hu et al., 2022), AdaLoRA(Zhang et al., 2023), DoRA(Liu et al., 2024c), VeRA(Kopiczko et al., 2024), and FourierFT(Gao et al., 2024), have demonstrated promising performance across various downstream tasks.

As shown in fig. 1, we discuss LoRA-style PEFT in this paper:

$$h_i = W_0 x_i + \mathrm{M}(x_i) \tag{1}$$

Here $W_0$ represents the pretrained weight, and the function $\mathrm{M}$ outputs the modification to $W_0 x_i$. The index $i$ indicates the position of token $x$ in the sequential input $X$. PEFT methods aim to reduce the number of trainable parameters within the function $\mathrm{M}$ and apply the modification to all $x_i$. This raises a natural question: Is it necessary to apply the modification on every position index?

Suppose that as research advances, we have developed a LM that can handle any potential downstream task with very high accuracy. Any further downstream fine-tuning would potentially degrade the performance on these tasks since the information contained within the LM would be altered. In this case, any PEFT modifications should not be applied on any position indices. As the capacity of current LMs increasing rapidly, doubt raises that how would the performance be affected if the modifications were applied selectively to a subset of position indices with one given downstream task sample.

In this paper, we aim to answer these questions by exploring a distinct pattern of modification, as shown in fig. 2:

$$h_i = \begin{cases} W_0 x_i & \text{if } S(W_0 x_i, M(x_i)) = 0 \\ W_0 x_i + M(x_i) & \text{if } S(W_0 x_i, M(x_i)) = 1 \end{cases} \tag{2}$$

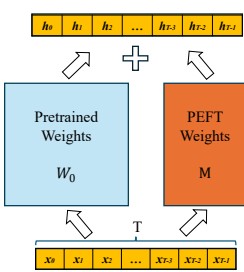

Figure 1: PEFT

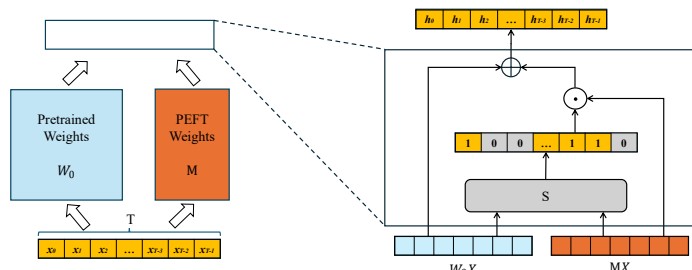

Figure 2: PEFT_S

Figure 3: Comparison of PEFT and PEFT_S

Function S determines a subset of position indices in which the PEFT modifications should be applied. If we can find such a function S that achieves similar or even better results on downstream tasks, we can demonstrate that it is unnecessary to apply the modification to all tokens in a layer during inference.

Our contributions are as follows:

- We select a simple but effective S and develop an efficient algorithm to train its scalar parameter along with the parameters of the PEFT methods. The whole algorithm is referred as PEFT_S.
- We figure out that by applying modifications on only a subset of the position indices, better results can be achieved on downstream tasks.
- We show that our algorithm can serve as an importance selector for PEFT target modules and that adding more target modules may harm downstream task performance.

## 2 RELATED WORK

### 2.1 LOW-RANK ADAPTATION

Low-Rank Adaptation (LoRA) and its derivatives like AdaLoRA have shown potential in reducing the number of fine-tuning parameters while maintaining model performance. LoRA (Hu et al., 2022) simulates weight changes by introducing two low-rank trainable matrices A and B, thus avoiding direct adjustment of the parameters of large pre-trained models. Once training is complete, these low-rank matrices can be merged with the original weights, introducing no additional computational burden during inference. As an extension of LoRA, AdaLoRA (Zhang et al., 2023) further enhances the flexibility and efficiency of fine-tuning by adaptively adjusting the scale of the low-rank matrices to meet the needs of different tasks. DoRA (Liu et al., 2024c) dynamically adjusts the update strategy of the low-rank matrix to adapt to different task requirements. VeRA (Kopiczko et al., 2024) utilizes orthogonalization techniques to optimize the representation of low-rank matrices.

### 2.2 SPARSITY

Sparsity-guided techniques improve the efficiency and interpretability of LLM. By reducing the number of active parameters and concentrating on the core components, these methods not only make models more efficient but also provide clearer insights into their decision-making processes. Mixture of experts (MOE) models, exemplified by Switch Transformers (Fedus et al., 2022) and Mistral 8x7B (Jiang et al., 2024), are sparsely-activated in order to maintain a consistent computational expense with outrageous numbers of parameters. Some recent works(Chen et al., 2024;

Liu et al., 2024b; Zadouri et al., 2024) have already combined MoE models with LoRA to further enhance performance. Another typical application of sparsity is model pruning. In order to maintain the model's optimal performance with less computation, these methods (Ma et al., 2023; Xu et al., 2024; Xia et al., 2024) focus on pruning less influential paramenters, while keeping those more important. (Tan et al., 2024) introduces sparsity-guided techniques, which are designed to offer a comprehensive interpretation of LLMs. Sparsity can be applied to various aspects of neural networks, including weights, activations, and gradients. While previous studies have predominantly focused on parameter-wise sparsity, our research delves into token(sequential)-wise sparsity.

## 3 THEORY

In this section, we present the theoretical derivation of our algorithm. Firstly, a simple but efficient trainable function S is introduced, which is designed to control the activation and deactivation of PEFT methods across sequential input $X$. Given that the gradient of parameters in S is 0 almost everywhere, proximal optimization technique is utilized to approximate the necessary updates. Finally, momentum calculations and adaptive learning rate are incorporated to be compatible with the widely-utilized Adam optimizer.

### 3.1 CONTROL FUNCTION S

**Brief Introduction to LoRA** Low-Rank Adaptation (LoRA) uses the product of two low rank matrices $A \in \mathbb{R}^{r \times k}$ and $B \in \mathbb{R}^{d \times r}$ to construct M:

$$\mathrm{M}(x_i) = BAx_i \tag{3}$$

where $r \ll \min(d, k)$ ensures that the number of parameters being trained is much smaller than the total number of parameters of the model. AdaLoRA, DoRA, and other methods explored in this paper use different ways to construct M to achieve better performance or increased efficiency. They are all state-of-the-art PEFT methods officially implemented in the popular Hugging Face library[1].

**Choice of Function** S We choose a simple and intuitive form of S:

$$\mathrm{S}(W_0, \mathrm{M}, x_i) = \begin{cases} 0 & \text{if } \|\mathrm{M}(x_i)\|_2 / \|W_0 x_i\|_2 < \tau \\ 1 & \text{if } \|\mathrm{M}(x_i)\|_2 / \|W_0 x_i\|_2 \geq \tau \end{cases} \tag{4}$$

where $\tau$ is a trainable scalar parameter for each target layer, optimized concurrently with the parameters of the PEFT methods. This choice of S discards the modification if the ratio of the modification's norm to the norm of the original output $W_0 x_i$ is below the threshold $\tau$.

Challenge lies in the training of the scalar parameter $\tau$, as S is a step function with respect to $\tau$, which complicates the application of standard backpropagation for computing the gradient of the objective function with respect to $\tau$. Furthermore, it is necessary to devise a mechanism ensuring that S permits only a subset of outputs, indexed by different positions, to be modified by M. To surmount these challenges, we propose a method that leverages proximal optimization techniques (Parikh et al., 2014) and employs an approximation of the gradient for the step function.

### 3.2 GRADIENT APPROXIMATE

**Proximal Optimization** In the context of Proximal Optimization methods, as delineated in (Parikh et al., 2014), each iteration of the optimization process involves minimizing the objective function augmented by a proximity term. This term serves to penalize deviations in the parameter updates, thereby ensuring that changes are constrained within a predefined proximity to the current solution. In our approach, we specifically target the parameter $\tau$ to incorporate a controlled modification mechanism, allowing partial modification of PEFT parameters.

---

[1]https://huggingface.co/

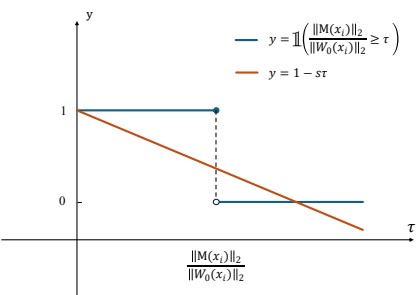

Figure 4: Gradient of $\tau$ can be approximated as $-s$

Given an objective function $L$, to elucidate the updating of parameter $\tau$ initially within the context of a stochastic gradient descent (SGD) framework, the update rule for $\tau$ at iteration $k+1$ is:

$$\tau^{k+1} = \operatorname*{argmin}_{\tau} \left[ \frac{1}{2\alpha_k} \left| \tau - \tau^k \right|^2 + (\tau - \tau^k) * \frac{\partial L}{\partial \tau^k} \right] \tag{5}$$

where $\alpha_k$ is the learning rate. A proximity term denoted as $P$ is added to eq. (5):

$$\tau^{k+1} = \operatorname*{argmin}_{\tau} \left[ \frac{1}{2\alpha_k} \left| \tau - \tau^k \right|^2 + (\tau - \tau^k) \cdot \frac{\partial L}{\partial \tau^k} \right.$$
$$\left. + \lambda \sum_{i=0}^{T-1} \mathbb{1} \left( \frac{\| \mathrm{M}(x_i) \|_2}{\| W_0 x_i \|_2} \geq \tau \right) \right] \tag{6}$$

where $T$ is the length of input tokens, $\lambda$ is the strength of the proximity term and $\mathbb{1}(\cdot)$ is the indicator function. This proximity term tends to enlarge $\tau$ when the modification $\mathrm{M}(x_i)$ is applied on many position indices.

To solve eq. (6), we need to compute $\frac{\partial L}{\partial \tau}$ and $\frac{\partial P}{\partial \tau}$. eq. (2) can be rewritten as

$$h_{ij} = [W_0 x_i]_j + [M(x_i)]_j * \mathbb{1} \left( \frac{\| \mathrm{M}(x_i) \|_2}{\| W_0 x_i \|_2} \geq \tau \right) \tag{7}$$

where $j \in [0, ..., C-1]$ and $C$ is the channel size. By backpropagation, we get:

$$\frac{\partial L}{\partial \tau} = \sum_{i=0}^{T-1} \mu_i * \frac{\partial \mathbb{1} \left( \frac{\| \mathrm{M}(x_i) \|_2}{\| W_0 x_i \|_2} \geq \tau \right)}{\partial \tau} \tag{8}$$

and

$$\frac{\partial P}{\partial \tau} = \sum_{i=0}^{T-1} \lambda * \frac{\partial \mathbb{1} \left( \frac{\| \mathrm{M}(x_i) \|_2}{\| W_0 x_i \|_2} \geq \tau \right)}{\partial \tau} \tag{9}$$

where

$$\mu_i = \sum_{j=0}^{C-1} \frac{\partial L}{\partial h_{ij}} * [M(x_i)]_j \tag{10}$$

**Approximate Gradient** From fig. 4 we observe that function $\mathbb{1} \left( \frac{\| \mathrm{M}(x_i) \|_2}{\| W_0 x_i \|_2} \geq \tau \right)$ decreases w.r.t $\tau$. Thus the partial gradient can be approximated as follows:

$$\frac{\partial \mathbb{1} \left( \frac{\| \mathrm{M}(x_i) \|_2}{\| W_0 x_i \|_2} \geq \tau \right)}{\partial \tau} = -s \tag{11}$$

where $s$ is a positive scalar hyperparameter. Given the coarse nature of this approximation, it is imperative to exercise caution when integrating eq. (11) into eq. (8) and eq. (9). Directly substituting

eq. (11) into eq. (8) and eq. (9) yields eq. (12) and eq. (13):

$$\frac{\partial L}{\partial \tau} = \sum_{i=0}^{T-1} \mu_i * (-s) \tag{12}$$

$$\frac{\partial P}{\partial \tau} = \sum_{i=0}^{T-1} \lambda * (-s) \tag{13}$$

eq. (13) always enlarges $\tau$, and the effect of eq. (12) depends on the sign of $\mu_i$. eq. (11) is quite a rough approximation and experimentally the usage of eq. (12) and eq. (13) leads to highly oscillatory updates of $\tau$ and poor model performance. Since fig. 4 shows that for certain value domains the approximate gradient w.r.t $\tau$ should be zero, we should add indicator conditions to constrain the effects of eq. (13) and eq. (12).

In the following we denote $\frac{\| M(x_i) \|_2}{\| W_0 x_i \|_2}$ as $r_i$. Extensive experiments show that the constrained variants of eq. (13) and eq. (12) (i.e., eq. (14) and eq. (15)) work well:

$$\frac{\partial L}{\partial \tau} = \sum_{i=0}^{T-1} \mathbb{1} \left[ \mathbb{1} \left( \mu_i \geq 0 \right) = \mathbb{1} \left( r_i \geq \tau \right) \right] * \mu_i * (-s) \tag{14}$$

$$\frac{\partial P}{\partial \tau} = \sum_{i=0}^{T-1} \mathbb{1} \left( r_i \geq \tau \right) * \lambda * (-s) \tag{15}$$

The constraints can be interpreted as follows: Whenever $\tau$ is either large enough ($\mu_i \geq 0$, $r_i < \tau$) or small enough ($\mu_i \leq 0$, $r_i \geq \tau$) to render the update of $\tau$ unnecessary, eq. (14) zeros $\frac{\partial L}{\partial \tau}$; similarly, when $\tau$ is large enough ($r_i < \tau$, $\lambda$ is a positive constant), eq. (15) nullifies $\frac{\partial P}{\partial \tau}$.

Given eq. (14) and eq. (15), the solution to eq. (6) is:

$$\tau^{k+1} = \tau^k + \alpha_k s * g_k \tag{16}$$

$$g_k = \sum_{i=0}^{T-1} \left\{ \mathbb{1} \left[ \mathbb{1} \left( \mu_i \geq 0 \right) = \mathbb{1} \left( r_i \geq \tau \right) \right] \mu_i + \mathbb{1} \left( r_i \geq \tau \right) \lambda \right\} \tag{17}$$

where $-\alpha_k s$ can be regarded as the learning rate, and $g_k$ can be considered the gradient of $\tau$.

## 3.3 ADAPTIVE MOMENT ESTIMATION

Adam(Kingma & Ba, 2015) has become one of the default practical choices of optimization techniques for deep learning due to its effective amalgamation of momentum and adaptive learning rate strategies within a singular, efficient framework. We implement the update of $\tau$ similarly to Adam:

$$m_k \leftarrow \beta_1 m_{k-1} + (1 - \beta_1) g_k \tag{18}$$

$$v_k \leftarrow \beta_2 v_{k-1} + (1 - \beta_2) g_k^2 \tag{19}$$

$$\hat{m}_k \leftarrow m_k / (1 - \beta_1^k) \tag{20}$$

$$\hat{v}_k \leftarrow v_k / (1 - \beta_2^k) \tag{21}$$

$$\tau^{k+1} \leftarrow \tau^k + \alpha_k s \frac{\hat{m}_k}{\sqrt{\hat{v}_k} + \epsilon} \tag{22}$$

where $\beta_1$ controls the contribution of past gradients, $\beta_2$ governs the decay rate of squared gradients, and $\epsilon$ is a minuscule constant to prevent division by zero.

## 3.4 OVERALL ALGORITHM

The proposed algorithm can be applied to any PEFT method expressible in the form presented by eq. (1). As depicted in the forward and backward steps outlined in algorithm 1, the parameter $\tau$ is updated simultaneously with the parameters in M. The PEFT method augmented with our algorithm is referred as PEFT_S, where S signifies the effect of PEFT_S is **S**equential **S**parse.

---

**Algorithm 1** Forward and Backward

---

1: **Input:** M, $\alpha_k$, $s$, $\lambda$, $\beta_1$, $\beta_2$, $\tau^k$
2: **Output:** $\tau^{k+1}$, updated parameters of M
3: **Forward Pass:**
4:    Forward as in eq. (2)
5: **Backward Pass:**
6:    **Step 1:** Perform the usual backward step to update parameters of M
7:    **Step 2:**
8:       Compute $\mu_i$ as in eq. (10)
9:       Compute $g_k$ as in eq. (17)
10:      Update $\tau^{k+1}$ as in eq. (22)

---

Table 1: Evaluation scores of LLaMA3-8B on the commonsense reasoning tasks for each method, along with sparsity and hyperparameters $s$ and $\lambda$ for each PEFT_S counterpart.

| | PIQA | BoolQ | HellaSwag | WinoGrande | SIQA | OBQA | ARC-e | ARC-c | Avg |
|---|---|---|---|---|---|---|---|---|---|
| LoRA | 88.5 | 64.2 | 95.3 | **85.5** | 80.8 | 85.4 | **90.3** | **79.9** | 83.7 |
| LoRA_S | **88.6** | **70.1** | **95.5** | 84.4 | **82.3** | **85.8** | 90.1 | 79.4 | **84.5** |
| sparsity(%) | 58.8 | 55.6 | 57.1 | 54.0 | 54.9 | 56.8 | 59.2 | 60.0 | 57.1 |
| $s$ | | | | 4e-5 | | | | | |
| $\lambda$ | | | | 4.5e-5 | | | | | |
| DoRA | 87.5 | 75.0 | 95.5 | 85.5 | 80.1 | 85.8 | 90.8 | 79.5 | 85.0 |
| DoRA_S | **88.8** | **75.2** | 95.5 | **87.1** | **80.2** | **86.6** | **91.2** | **80.1** | **85.6** |
| sparsity(%) | 50.0 | 47.9 | 50.2 | 47.1 | 47.6 | 48.3 | 49.6 | 50.0 | 48.8 |
| $s$ | | | | 4e-5 | | | | | |
| $\lambda$ | | | | 1e-5 | | | | | |
| AdaLoRA | **88.8** | 74.2 | 95.5 | 85.2 | 79.6 | 85.4 | **91.2** | 79.5 | 84.9 |
| AdaLoRA_S | 88.4 | **75.5** | **95.8** | **85.6** | **80.9** | **86.2** | 90.1 | **79.9** | **85.3** |
| sparsity(%) | 74.3 | 76.0 | 65.0 | 70.8 | 68.9 | 72.8 | 75.5 | 75.9 | 67.7 |
| $s$ | | | | 4e-5 | | | | | |
| $\lambda$ | | | | 1e-4 | | | | | |

## 4 EXPERIMENTS

We have conducted a comprehensive set of experiments to ascertain the efficacy of PEFT across various position indices within a neural network layer. Our findings indicate that the indiscriminate application of PEFT to all indices is not only superfluous but may also be counterproductive. A judicious selection of PEFT modifications can significantly bolster the performance on subsequent tasks. Throughout these experiments, we set $\alpha = 1$, $\beta_1 = 0.9$, $\beta_2 = 0.98$. All primary experiments were conducted at least three times with different seeds, and the average scores were reported. The detailed hyperparameters can be found in section A.4.

### 4.1 COMMONSENSE REASONING

We evaluate the performance of LoRA, DoRA, and AdaLoRA, as well as their PEFT_S counterparts LoRA_S, DoRA_S, and AdaLoRA_S, on eight sub-tasks of the commonsense reasoning task (Liu et al., 2024c). We use LLaMA3-8B (Dubey et al., 2024) as the base model and combine the training datasets from all eight tasks into a unified dataset for final training (Hu et al., 2023). Evaluations are conducted separately on each test dataset. We set the rank of LoRA and DoRA to 32. For AdaLoRA, we set the initial rank to 64 and the target rank to 32 for fairness. We use the same rank for LoRA_S, DoRA_S and AdaLoRA_S.

table 1 shows the evaluation scores of each PEFT method, as well as the sparsity and hyperparameters $s$ and $\lambda$ for the PEFT_S counterparts. eq. (23) provides the definition of sparsity for a single target module and data sample. Sparsity indicates the number of position indices of the input se-

Table 2: Evaluation scores of LLaVA-1.5-7B on the visual instruction tuning tasks for each method, along with sparsity and hyperparameters $s$ and $\lambda$ for each PEFT_S counterpart.

| | SQA | POPE | MMBench | GQA | VisWiz | VQAv2 | VQAT | Avg |
|---|---|---|---|---|---|---|---|---|
| LoRA | 68.1 | 87.3 | **64.8** | 63.1 | 46.8 | **79.1** | 57.3 | 66.6 |
| LoRA_S | **68.3** | **87.5** | 63.6 | **63.3** | **50.4** | 78.3 | **57.5** | **67.0** |
| sparsity(%) | 67.6 | 59.3 | 62.0 | 59.3 | 59.5 | 59.2 | 62.2 | 61.3 |
| $s$ | | | | 4e-5 | | | | |
| $\lambda$ | | | | 2e-6 | | | | |
| DoRA | 67.5 | 87.4 | **65.5** | **63.0** | 50.5 | **78.6** | **57.0** | 67.1 |
| DoRA_S | **69.4** | **88.0** | 64.7 | 62.4 | **54.2** | 77.8 | 55.9 | **67.4** |
| sparsity(%) | 70.6 | 62.6 | 65.2 | 62.4 | 62.9 | 63.8 | 65.2 | 64.7 |
| $s$ | | | | 4e-5 | | | | |
| $\lambda$ | | | | 1e-6 | | | | |
| AdaLoRA | **68.0** | 87.3 | **64.3** | 61.7 | 48.9 | 78.3 | **56.9** | 66.5 |
| AdaLoRA_S | 67.8 | **87.5** | 63.4 | **62.3** | **52.6** | **78.5** | 55.2 | **66.8** |
| sparsity(%) | 60.0 | 50.9 | 54.1 | 50.9 | 51.4 | 50.9 | 53.9 | 53.2 |
| $s$ | | | | 4e-5 | | | | |
| $\lambda$ | | | | 5e-7 | | | | |

quence that are not modified by the outputs of PEFT methods. We present the average sparsity across all target modules and data samples in table 1.

$$\text{sparsity}(W_0, \text{M}, X) = 1 - \frac{1}{T} \sum_{i=0}^{T-1} \text{S}(W_0 x_i, \text{M}(x_i)) \tag{23}$$

When applied with our algorithm, the average score of each PEFT method improves. With an average sparsity of $57.1\%$, LoRA_S surpasses LoRA by an absolute value of 0.8. With an average sparsity of $48.8\%$, DoRA_S exceeds DoRA by an absolute value of 0.6. With a higher average sparsity of $67.7\%$, AdaLoRA_S surpasses AdaLoRA by an absolute value of 0.4. This demonstrates that by applying the modifications of PEFT methods to only half of the position indices of the input sequence during inference, we can achieve even better performance.

## 4.2 VISUAL INSTRUCTION TUNING

We also conduct experiments on visual instruction tuning tasks, which are generative language tasks that contain open-ended questions about images, using LLaVA-1.5-7B (Liu et al., 2024a; 2023) as the base model. We adopt a training setup similar to that in (Liu et al., 2024c). The rank of LoRA and DoRA is set to 128, while the initial and target ranks of AdaLoRA are set to 256 and 128, respectively. The same rank is used for LoRA_S, DoRA_S, and AdaLoRA_S.

table 2 shows the results. The average score of each PEFT method improves when applied with our algorithm. With an average sparsity of $61.3\%$, LoRA_S exceeds LoRA by an absolute value of 0.4. With an average sparsity of $64.7\%$, DoRA_S exceeds DoRA by an absolute value of 0.3. With an average sparsity of $53.2\%$, AdaLoRA_S exceeds AdaLoRA by an absolute value of 0.3. This experiment further demonstrates that better performance is achieved when the modification of PEFT is applied to less than half of the position indices of the input sequence during inference.

## 4.3 NATURAL LANGUAGE UNDERSTANDING

We also conduct experiments on tasks from the General Language Understanding Evaluation (GLUE) benchmark (Wang et al., 2019) using DeBERTaV3-base (He et al., 2023) as the base model. The target rank of LoRA and AdaLoRA is set to 4, and the hidden size of the Parallel Adapter (Hu et al., 2023) is set to 32 for fair comparison. The results are shown in table 3. With an average sparsity of approximately $60\%$, PEFT_S matches or exceeds the average performance of its corresponding PEFT method.

Table 3: Evaluation scores of DeBERTaV3-base on the GLUE benchmark tasks for each method, along with sparsity and hyperparameters $s$ and $\lambda$ for each PEFT_S counterpart.

| | MNLI | CoLA | QNLI | RTE | MRPC | QQP | SST-2 | STS-B | Avg |
|---|---|---|---|---|---|---|---|---|---|
| LoRA | 89.9 | 69.2 | 94.1 | 87.1 | 90.4 | **92.2** | 95.3 | **91.8** | 88.8 |
| LoRA_S | **89.9** | **70.3** | **94.2** | **88.1** | **91.2** | 92.0 | **95.8** | 91.5 | **89.1** |
| sparsity(%) | 71.6 | 64.1 | 70.6 | 42.8 | 55.4 | 57.1 | 67.6 | 33.9 | 57.9 |
| $s$ | | | | | 1e-4 | | | | |
| $\lambda$ | 1e-5 | 6e-6 | 2e-5 | 1e-5 | 1e-5 | 2e-6 | 2e-5 | 8e-5 | |
| AdaLoRA | 90.4 | 71.1 | **94.6** | 88.6 | 91.3 | 91.9 | 95.9 | **91.9** | 89.5 |
| AdaLoRA_S | **90.5** | **71.3** | 94.5 | **89.0** | **91.6** | 91.9 | **96.1** | 91.6 | **89.6** |
| sparsity(%) | 55.2 | 53.1 | 68.1 | 62.7 | 71.7 | 54.7 | 56.9 | 68.3 | 61.3 |
| $s$ | | | | | 1e-4 | | | | |
| $\lambda$ | 9e-6 | 5e-6 | 4.5e-7 | 2e-6 | 2e-8 | 5e-7 | 2e-6 | 5e-6 | |
| Adapter | **90.4** | **72.1** | **94.2** | 86.6 | 90.0 | **92.1** | 95.6 | **91.4** | 89.1 |
| Adapter_S | 90.1 | 72.0 | 94.1 | **86.8** | **91.4** | 91.8 | **96.6** | 91.3 | **89.3** |
| sparsity(%) | 56.4 | 59.4 | 65.5 | 62.7 | 55.1 | 58.0 | 56.2 | 67.7 | 60.1 |
| $s$ | | | | | 1e-4 | | | | |
| $\lambda$ | 5e-7 | 4.5e-7 | 4.5e-7 | 2e-6 | 2e-8 | 5e-7 | 2e-6 | 5e-6 | |

# 5  IN-DEPTH ANALYSIS

Table 4: The performance of LLaMA3-8B on the commonsense reasoning task using various selection methods.

| | PIQA | BoolQ | HellaSwag | WinoGrande | SIQA | OBQA | ARC-e | ARC-c | Avg |
|---|---|---|---|---|---|---|---|---|---|
| S_low_50% | **88.6** | **75.7** | 95.4 | **86.9** | 80.6 | **87.6** | **90.8** | 79.7 | **85.7** |
| S_high_50% | 88.2 | 62.2 | 92.2 | 85.2 | 80.2 | 86.2 | 90.4 | 79.9 | 83.0 |
| norm_relative_50% | 88.4 | 62.2 | 95.4 | 85.0 | 80.5 | 83.6 | 90.1 | **80.2** | 83.2 |
| norm_abs_50% | 88.4 | 52.7 | 85.8 | 83.9 | 78.7 | 86.2 | 89.6 | 79.0 | 80.5 |
| random_50% | 88.3 | 73.4 | 91.2 | 85.6 | 80.6 | 85.7 | 89.6 | 78.9 | 84.2 |
| half_rank_50% | 88.1 | 75.4 | 95.2 | 86.3 | **81.2** | 85.2 | 89.6 | 78.7 | 84.9 |

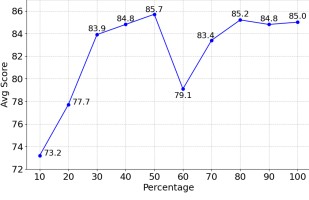

Figure 5: DoRA

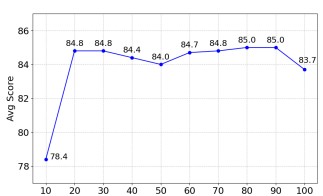

Figure 6: LoRA

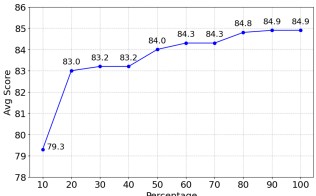

Figure 7: AdaLoRA

Figure 8: The performance of LLaMA3-8B using various PEFT methods on the commonsense reasoning task with different percentages of selected modules.

**The Necessity of the Adaptive Moment Estimation Module**  The Adaptive Moment Estimation module is indispensable: without it, $\tau$ oscillates and diverges, and the training loss fails to converge, confirming its stabilizing role (see section A.2).

**Module-wise Sparsity**  LLaMA3-8B has 32 layers; we averaged the sparsity of q_proj, k_proj, and v_proj in each layer and observed distinct patterns across PEFT methods (details in the section A.3).

For LoRA_S and AdaLoRA_S, sparsity values are distributed more evenly across modules than in DoRA_S. The standard deviation of sparsity values is 9.8 for LoRA_S, 11.4 for AdaLoRA_S, and 41.3 for DoRA_S. The main difference between LoRA and AdaLoRA is that AdaLoRA adjusts the

rank of weight matrices during LoRA-alike fine-tuning, although they still share the same weight structure. DoRA, on the other hand, constructs weights differently by decomposing the weight matrix into two separate components: magnitude and direction. We believe it is this difference in weight structure that causes the varying behaviors.

We found that many sparsity values of DoRA_S are close to either 100 or 0, indicating that some modules allow nearly all PEFT modifications, while others reject almost all. This raises a natural question: Are the modules with lower sparsity more important for PEFT on the downstream task?

**Importance Selection** We designed the following experiments to address this question:

- S_low_50%: We select the 50% of modules with the lowest average sparsity in DoRA_S fine-tuning as the target modules and fine-tune LLaMA3-8B with DoRA.

- S_high_50%: We select the 50% of modules with the highest average sparsity in DoRA_S fine-tuning as the target modules and fine-tune LLaMA3-8B with DoRA.

- norm_relative_50%: We select the 50% of modules with the highest average value of $\| \mathrm{M}(x_i)\|_2/\|W_0 x_i\|_2$ in DoRA fine-tuning as the target modules and fine-tune LLaMA3-8B with DoRA.

- norm_abs_50%: We select the 50% of modules with the highest average value of $\| \mathrm{M}(x_i)\|_2$ in DoRA fine-tuning as the target modules and fine-tune LLaMA3-8B with DoRA.

- random_50%: We randomly select 50% of modules as the target modules and fine-tune LLaMA3-8B with DoRA, performing the randomization three times.

- half_rank_50%: We use all modules as the target modules but with half the rank to maintain the same number of parameters as the above experiments.

table 4 shows the results. S_low_50% significantly outperforms S_high_50%, even though they contain the same number of trainable parameters. S_low_50% even exceeds the performance of DoRA with 100% trainable parameters (Avg 85.0, see table 1). The low performance of norm_relative_50% and norm_abs_50% indicates that the average value of $| \mathrm{M}(x_i)|_2/|W_0 x_i|_2$ or $| \mathrm{M}(x_i)|_2$ in DoRA fine-tuning cannot be used as a reliable importance selection indicator. The performance of random_50% and half_rank_50% further demonstrates that the superior performance of S_low_50% is not due to randomness or a reduction in the number of parameters. These experiments indicate that sparsity is a valid measure of the importance of target modules.

**More Target Modules May Lead to Performance Degradation** Using sparsity as a tool for importance selection, we address the question: Do more target modules lead to better performance? We select different percentages of modules, ranked in descending order by sparsity values, as the target modules and fine-tune LLaMA3-8B with PEFT. The results are shown in fig. 8. For AdaLoRA, performance does not decrease as the percentage of target modules increases. However, for DoRA and LoRA, selecting more target modules can lead to worse performance. For DoRA, the best performance is achieved when only 50% of modules are selected. For LoRA, the best performance is achieved when 80% of modules are selected, and a similar performance can be achieved with only 20% of modules.

We also observed a sudden drop in performance. When the percentage of target modules increases from 50% to 60%, the performance of DoRA decreases significantly by 6.6 points. Similarly, when the percentage of target modules increases from 90% to 100%, the performance of LoRA drops by 1.3 points. This suggests that joint training with less important target modules may negatively affect performance. AdaLoRA, on the other hand, is not adversely affected by less important target modules, as it can dynamically allocate ranks among target modules during fine-tuning.

## 6 CONCLUSION

We introduced a novel function denoted as S and propose an algorithm PEFT_S, which integrates the optimization of this scalar function with the existing PEFT parameters. Our approach enhances the performance of PEFT on downstream tasks, particularly when applied only to a subset of position indices. The algorithm serves as a discriminative importance filter for PEFT modules. Based on these insights, we outline several future research directions that could leverage PEFT_S as a foundational tool, advancing the optimization and application of PEFT methods.

# 7 REPRODUCIBILITY STATEMENT

To facilitate reproduction of the results reported in this paper, we provide a complete list of experimental hyper-parameters in section A.4 and the source-code implementation of PEFT_S in the supplementary materials.

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

## A APPENDIX

### A.1 LLM USAGE STATEMENT

During the preparation of this manuscript, the authors utilized a large language model (LLM) exclusively for grammar and language polishing. All technical content, scientific claims, and experimental results were conceived, derived, and verified solely by the authors.

### A.2 THE TRAINING LOSS PROGRESSION WITH AND WITHOUT THE ADAPTIVE MOMENT ESTIMATION

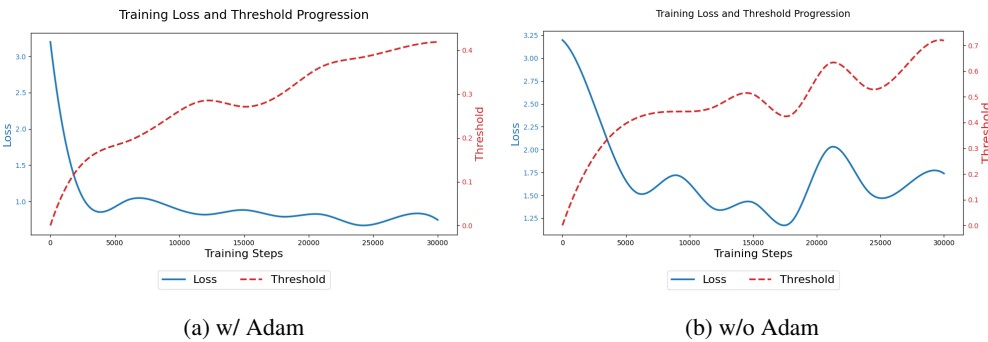

(a) w/ Adam                    (b) w/o Adam

Figure 9: Training loss progression for layer base_model.layers.20.self_attn.v_proj of LLaMA3-8B

### A.3 MODULE-WISE AVERAGE SPARSITY OF LLaMA3-8B ON THE COMMONSENSE REASONING TASK

### A.4 HYPERPARAMETERS

For the Commonsense Reasoning task, The hyperparameter $s$ was set to a constant value of 4e-5. The hyperparameter $\lambda$ was selected from the interval [1e-7, 1e-3] to identify the optimal parameter. table 6 shows the other hyperparameters.

For the Visual Instruction Tuning task, The hyperparameter $s$ was set to a constant value of 4e-5. The hyperparameter $\lambda$ was selected from the interval [1e-7, 1e-3] to identify the optimal parameter. table 7 shows the other hyperparameters.

For the Natural Language Understanding task, The hyperparameter $s$ was set to a constant value of 1e-4. The hyperparameter $\lambda$ was selected from the interval [1e-8, 1e-3] to identify the optimal parameter. table 8 shows the other hyperparameters.

Table 5: Module-wise average sparsity (%) of LLaMA3-8B on commonsense reasoning. "-" denotes rank 0.

| Layer | LoRA_S q_proj | k_proj | v_proj | avg | DoRA_S q_proj | k_proj | v_proj | avg | AdaLoRA_S q_proj | k_proj | v_proj | avg |
|---|---|---|---|---|---|---|---|---|---|---|---|---|
| 0 | 65.6 | 77.6 | 0.0 | 47.7 | 100.0 | 0.0 | 0.0 | 33.3 | - | - | 76.9 | 76.9 |
| 1 | 59.7 | 65.5 | 54.8 | 60.0 | 100.0 | 0.0 | 65.0 | 55.0 | 62.3 | 84.5 | 68.4 | 71.8 |
| 2 | 62.8 | 64.9 | 51.5 | 59.7 | 100.0 | 100.0 | 57.2 | 85.7 | 57.3 | 75.0 | 81.7 | 71.3 |
| 3 | 60.1 | 64.4 | 56.3 | 60.2 | 99.8 | 100.0 | 37.1 | 79.0 | 67.6 | 68.7 | 81.9 | 72.7 |
| 4 | 63.7 | 67.6 | 56.6 | 62.6 | 24.6 | 0.0 | 83.0 | 35.9 | 48.2 | 70.2 | 75.8 | 64.7 |
| 5 | 57.9 | 62.6 | 54.8 | 58.4 | 0.0 | 0.0 | 30.9 | 10.3 | 49.6 | 86.4 | 80.9 | 72.3 |
| 6 | 62.4 | 60.6 | 48.4 | 57.1 | 100.0 | 0.0 | 62.9 | 54.3 | 70.7 | 84.8 | 66.1 | 73.9 |
| 7 | 61.3 | 65.9 | 51.2 | 59.5 | 0.0 | 0.0 | 82.8 | 27.6 | 59.8 | 86.6 | 77.2 | 74.5 |
| 8 | 60.8 | 65.8 | 50.4 | 59.0 | 70.5 | 57.1 | 60.2 | 62.6 | 54.0 | 93.6 | 60.7 | 69.4 |
| 9 | 65.9 | 51.1 | 53.1 | 56.7 | 0.0 | 11.6 | 47.4 | 19.7 | 52.0 | 72.4 | 61.2 | 61.9 |
| 10 | 57.7 | 58.7 | 52.8 | 56.4 | 0.0 | 0.0 | 41.3 | 13.8 | 51.3 | 82.5 | 64.2 | 66.0 |
| 11 | 58.5 | 57.5 | 42.3 | 52.8 | 91.7 | 77.2 | 76.1 | 81.7 | 54.2 | 86.6 | 65.0 | 68.6 |
| 12 | 51.9 | 68.2 | 49.9 | 56.7 | 0.0 | 0.0 | 2.1 | 0.7 | 50.7 | 87.9 | 56.0 | 64.9 |
| 13 | 58.9 | 68.3 | 51.2 | 59.5 | 90.2 | 90.0 | 35.1 | 71.7 | 61.3 | 73.4 | 56.1 | 63.6 |
| 14 | 57.1 | 60.6 | 48.9 | 55.6 | 0.0 | 82.6 | 44.6 | 42.4 | 62.0 | 68.6 | 44.8 | 58.5 |
| 15 | 63.2 | 63.5 | 51.0 | 59.3 | 81.2 | 0.0 | 35.2 | 38.8 | 66.3 | 69.0 | 46.0 | 60.4 |
| 16 | 58.5 | 69.9 | 55.9 | 61.4 | 99.8 | 100.0 | 3.6 | 67.8 | 59.2 | 86.9 | 48.8 | 65.0 |
| 17 | 62.9 | 70.8 | 54.7 | 62.8 | 99.3 | 100.0 | 30.9 | 76.7 | 61.6 | 74.0 | 53.6 | 63.1 |
| 18 | 67.3 | 75.9 | 61.3 | 68.2 | 99.5 | 70.0 | 47.0 | 72.2 | 62.6 | 94.7 | 62.5 | 73.3 |
| 19 | 65.6 | 64.4 | 54.0 | 61.3 | 98.0 | 0.0 | 78.5 | 58.8 | 66.9 | 80.5 | 62.7 | 70.0 |
| 20 | 67.8 | 66.0 | 47.5 | 60.4 | 97.6 | 0.0 | 0.8 | 32.8 | 65.5 | 85.8 | 54.7 | 68.6 |
| 21 | 58.2 | 63.5 | 44.2 | 55.3 | 99.4 | 100.0 | 42.4 | 80.6 | 61.0 | 69.1 | 45.4 | 58.5 |
| 22 | 62.8 | 57.6 | 58.2 | 59.5 | 99.8 | 98.0 | 79.5 | 92.4 | 64.5 | 70.8 | 59.7 | 65.0 |
| 23 | 70.0 | 71.7 | 55.6 | 65.8 | 99.5 | 2.5 | 14.0 | 38.7 | 69.6 | 71.9 | 57.7 | 66.4 |
| 24 | 72.3 | 63.0 | 53.5 | 62.9 | 0.0 | 0.3 | 46.0 | 15.4 | 68.8 | 73.7 | 67.2 | 69.9 |
| 25 | 69.8 | 63.2 | 56.3 | 63.1 | 0.2 | 0.0 | 4.7 | 1.6 | 58.6 | 65.8 | 76.9 | 67.1 |
| 26 | 65.1 | 70.2 | 56.4 | 63.9 | 9.4 | 94.2 | 56.5 | 53.4 | 64.4 | 74.5 | 67.7 | 68.8 |
| 27 | 68.9 | 67.2 | 68.2 | 68.1 | 99.8 | 95.1 | 45.1 | 80.0 | 62.7 | 85.8 | 78.3 | 75.6 |
| 28 | 57.8 | 58.9 | 61.5 | 59.4 | 17.9 | 99.9 | 94.9 | 70.9 | 63.8 | 76.2 | 86.1 | 75.4 |
| 29 | 71.6 | 78.7 | 49.3 | 66.5 | 0.0 | 100.0 | 11.4 | 37.1 | 63.9 | 74.4 | 79.7 | 72.7 |
| 30 | 59.5 | 59.6 | 57.2 | 58.8 | 0.0 | 100.0 | 39.2 | 46.4 | 67.8 | 68.6 | 72.9 | 69.8 |
| 31 | 64.4 | 67.3 | 46.1 | 59.3 | 100.0 | 0.0 | 1.2 | 33.7 | 69.3 | 71.0 | 57.7 | 66.0 |
| avg | 64.8 | 67.4 | 53.3 | — | 60.6 | 47.7 | 43.8 | — | 61.2 | 77.9 | 67.6 | — |

Table 6: Hyper-parameter configuration for commonsense reasoning.

| Rank | Scale | Dropout | Optimizer | LR | Scheduler | Batch | Epochs | Target |
|---|---|---|---|---|---|---|---|---|
| 32 | 0.5 | 0.05 | AdamW | 1e-4 | Linear | 16 | 3 | Q,K,V,Up,Down |

Table 7: Hyper-parameter configuration for visual instruction tuning.

| Rank | Scale | Dropout | Optimizer | LR | Scheduler | Batch | Epochs | Target |
|---|---|---|---|---|---|---|---|---|
| 128 | 0.5 | 0.05 | AdamW | 2e-4 | Cosine | 16 | 1 | Q,K,V,O,Up,Down,Gate |

Table 8: Hyperparameter configurations used for the Natural Language Understanding task.

| | Rank | Scaling | Dropout | Optimizer | LR | Scheduler | Batch | Epochs | Target |
|---|---|---|---|---|---|---|---|---|---|
| CoLA | 4 | 0.5 | 0.1 | AdamW | 4e-4 | Linear | 32 | 40 | Q,K,V,O,I |
| MNLI | 4 | 0.5 | 0.15 | AdamW | 2.5e-4 | Linear | 32 | 10 | Q,K,V,O,I |
| QNLI | 4 | 0.5 | 0.1 | AdamW | 2e-4 | Linear | 32 | 10 | Q,K,V,O,I |
| RTE | 4 | 0.5 | 0.2 | AdamW | 1e-4 | Linear | 32 | 80 | Q,K,V,O,I |
| MRPC | 4 | 0.5 | 0 | AdamW | 1e-4 | Linear | 32 | 60 | Q,K,V,O,I |
| QQP | 4 | 0.5 | 0.15 | AdamW | 4e-4 | Linear | 32 | 10 | Q,K,V,O,I |
| SST-2 | 4 | 0.5 | 0 | AdamW | 4e-4 | Linear | 32 | 48 | Q,K,V,O,I |
| STS-B | 4 | 0.5 | 0.2 | AdamW | 5e-4 | Linear | 32 | 45 | Q,K,V,O,I |

