# OpenReview forum: "IS IT NECESSARY TO APPLY PARAMETER-EFFICIENT FINE-TUNING TO ALL POSITION INDICES?"
_ICLR.cc/2026/Conference — ICLR 2026 Conference Withdrawn Submission_

### Official Review · Reviewer_r9GL · 2025-10-30

**Soundness:** 2
**Presentation:** 2
**Contribution:** 3
**Rating:** 6
**Confidence:** 4

**Summary:**

This submission introduces a new LoRA variant, PEFT_S, which incorporates a trainable gating mechanism to selectively apply LoRA modifications to only a subset of tokens of a sequence. The core of this mechanism is a simple, trainable threshold, $\tau$, which determines whether to apply a modification based on the relative norm of the LoRA update compared to the original output. Through extensive experiments on commonsense reasoning, visual instruction tuning, and natural language understanding tasks, the submission demonstrates that this token-wise selective LoRA can lead to better performance than the standard LoRA. Furthermore, the analysis reveals that learned sparsity can serve as an indicator of module importance, suggesting that targeting fewer modules can be more effective.

**Strengths:**

- Interesting idea: The investigation into whether LoRA modifications are needed at every token position is an interesting direction that has not been explored much.
- Good Empirical Results: The experimental results are comprehensive and compelling. PEFT_S consistently improves or matches the performance of mentioned baselines (LoRA, DoRA, AdaLoRA) across a diverse set of models (LLaMA3, LLaVA, DeBERTaV3) and tasks. The fact that these gains are achieved with significant sparsity (often over 50%) makes the results particularly impressive and practically relevant.
- Interesting analysis: The finding that fine-tuning with only the 50% "most important" modules (as identified by PEFT_S) can outperform fine-tuning with all modules is an interesting result that could have a significant impact on how practitioners approach PEFT. It provides a data-driven method for optimizing the application of PEFT.

**Weaknesses:**

- Hyperparameter Sensitivity: The method introduces extra hyperparameters, $s$ and $\lambda$, which appear crucial for successful training. The paper reports the values used but does not include an ablation study or discussion on their sensitivity. This leaves open the question of how difficult it is to tune these parameters for a new model or dataset.
- Clarity: The connection between the proximal optimization framework and the final, specific update rule feels abrupt. A more detailed, step-by-step motivation for why this particular form of constrained gradient approximation was chosen will strengthen the clarity.

**Questions:**

- How sensitive is the final performance and learned sparsity to $s$ and $\lambda$? Could you provide a brief ablation study or discuss the general strategy for tuning them?
- How does the rank of LoRA affect the final performance?
- How about the performance compared against similar baselines, such as LoRA-MoE[1] and HydraLoRA[2]? Why not take them as baselines?
- Some analysis[3-5] about the PEFT can be considered as a reference.

[1] LoRAMoE: Alleviate World Knowledge Forgetting in Large Language Models via MoE-Style Plugin

[2] HydraLoRA: An Asymmetric LoRA Architecture for Efficient Fine-Tuning

[3] An Empirical Study of Parameter Efficient Fine-tuning on Vision-Language Pre-train Model

[4] Benchmarking Robustness of Adaptation Methods on Pre-trained Vision-Language Models

[5] Revisiting Parameter-Efficient Tuning: Are We Really There Yet?

---

### Official Review · Reviewer_eEVp · 2025-10-31

**Soundness:** 2
**Presentation:** 2
**Contribution:** 3
**Rating:** 6
**Confidence:** 3

**Summary:**

This submission proposes PEFT_S, which utilizes a threshold gating mechanism to selectively apply LoRA modifications for a subset of tokens. To achieve this, the PEFT_S introduces proximal optimization to train a threshold, $\tau$, which selects a part of tokens to apply LoRA modifications. The extensive experiments on commonsense reasoning, visual instruction tuning, and natural language understanding tasks across three LLMs demonstrate the superior performance of PEFT_S against vanilla LoRA and other baselines. The analysis results also indicate that such partial token PEFT is effective.

**Strengths:**

- The paper introduces a simple yet effective trainable gating mechanism to increase the sparsity of LoRA.
- The paper proposes a proximal-gradient approximation to handle the instability of the trainable threshold $tau$.
- The experimental results demonstrate the performance gains on eight commonsense reasoning tasks, four visual-QA benchmarks, and eight GLUE tasks. And the ablations on selecting low- vs. high-sparsity modules confirm that sparsity of PEFT is effective.

**Weaknesses:**

- This submission only proposes a method to improve the performance of LoRA and does not generalize to other PEFT methods, such as Adapter tuning, and so on. Thus, the use of the description of PEFT is somewhat overclaimed.
- It is unclear under what conditions the proximal gradient stabilizes the optimization of $tau$ and how to choose $s$ and $\lambda$.
- Sparsity suggests reduced compute. No actual latency or FLOP results are provided to confirm practical speedups.

**Questions:**

- What is the difference between PEFT_S and LoRA+MoE[1]?
- The reason to use proximal optimization for $tau$ is unknown. Why use it? What are the pros compared to other optimization methods, such as the EM algorithm?
- How about the computation cost? Is it greater than LoRA?
- How about the inference time of PEFT_S? Does it still outperform the baselines?
- Some related works[2] can be considered as a reference.

[1] PERFT: Parameter-Efficient Routed Fine-Tuning for Mixture-of-Expert Model

[2] Parameter Efficient Multi-task Fine-tuning by Learning to Transfer Token-wise Prompts

---

### Official Review · Reviewer_xbqE · 2025-11-01

**Soundness:** 3
**Presentation:** 2
**Contribution:** 4
**Rating:** 2
**Confidence:** 5

**Summary:**

This paper addresses a limitation of LoRA-based parameter-efficient fine-tuning (PEFT): not all position indices need to be updated. To mitigate redundant updates, the authors introduce a learnable threshold $\tau$ that selectively activates LoRA modules on important token positions. $\tau$ is optimized via proximal optimization (Parikh and Boyd, 2014), which stabilizes training of the non-differentiable step function. Experiments on commonsense reasoning, visual instruction tuning tasks, and GLUE demonstrate substantial performance improvements.

**Strengths:**

1. The proposed method is novel and well-motivated, introducing proximal optimization into LoRA-based fine-tuning for selective position-wise updates.
2. Experiments across multiple benchmarks demonstrate consistent and sometimes substantial performance improvements over baseline PEFT methods.

**Weaknesses:**

1. The scalar $s$ in Eq.(11) is treated as a hyperparameter, yet the paper provides no theoretical justification or tuning analysis for its chosen values.
2. The set of baselines is inconsistent across benchmarks: For example, Table 3 (GLUE) includes Adapter for comparison, while Tables 1–2 (Commonsense Reasoning and Visual Instruction Tuning) only compare LoRA-based variants, making it difficult to assess performance improvements under a unified experimental setting.
3. The authors did not discuss why Adapter is better than Adapter_S on several datasets in Table 3.
4. Some writing flaws must be fixed:
- The paper does not explicitly define the proximity term $P$ in Eq.(6); readers must guess its form from the context.
- The variable $\mu_i$ in Eq.(10) is introduced without explanation, making the gradient derivation harder to follow.
- The caption of Figure 4 is too brief and does not clearly describe what the figure shows or how it relates to Eq.(11).
- Some symbols in Section 3.3 (e.g., $m_k$, $v_k$) are in a lack of introduction.

**Questions:**

1. How to choose the best values of $s$ and $\lambda$?
2. Is there any reason using Frobenius norm in Eq.(4)?

---

### Official Review · Reviewer_2iKT · 2025-11-01

**Soundness:** 2
**Presentation:** 2
**Contribution:** 2
**Rating:** 4
**Confidence:** 4

**Summary:**

This paper proposes PEFT-S, a selective fine-tuning method that applies LoRA-style parameter updates only to a subset of token positions, rather than all indices. The authors introduce a trainable threshold function, S, that determines whether each position should receive a PEFT modification, optimized jointly with model parameters via proximal optimization and an Adam-like adaptive moment estimation to stabilize training. Experiments on LLaMA-3-8B, DeBERTaV3, and LLaVA-1.5-7B show that fine-tuning only ~50–60 % of positions can match or slightly outperform full PEFT.

**Strengths:**

* Conceptually simple yet novel exploration of token-wise sparsity in PEFT
* Provides analytical formulation and solid experimental coverage across NLP and vision-language tasks
* Demonstrates empirical efficiency gains with reduced parameter updates

**Weaknesses:**

In your manuscript, you state: “Suppose that as research advances, we have developed an LM that can handle any potential downstream task with very high accuracy.” Did you actually develop such a language model? Was this a mistake made by ChatGPT? Please clarify this point.

The gradient approximation for τ (Eq. 11) is crude and heuristic, lacking rigorous justification. Sparse selection may overlap with existing pruning or gating effects, making causality unclear. Both your figures and sub-figures are labeled as “Figure X.” It would be better to improve the format.

Please evaluate your proposed method on NLG tasks and include additional experimental results and analyses to strengthen the empirical evidence of your work.
Overall, the proposed method lacks novelty, as the problem formulation and the proposed idea are not causally or logically well-connected, making it difficult to identify a strong, convincing contribution.

**Questions:**

You write: “Any further downstream fine-tuning would potentially degrade the performance on these tasks since the information contained within the LM would be altered.” Please provide justification or supporting evidence for this claim. Based on your current reasoning, PEFT methods would perform worse than in-context learning. If you intend to refer to more complex capabilities such as instruction following or chain-of-thought generation, the issue is that no corresponding experiments or analyses are provided to justify your statement.

---

### Note · Authors · 2025-11-25

I have read and agree with the venue's withdrawal policy on behalf of myself and my co-authors.